



# Measurement Report: Determination of aerosol vertical features on different time-scales over East Asia based on CATS aerosol products

Yueming Cheng[1, 2], Tie Dai[1, 2*], Jiming Li[3], Guangyu Shi[1, 2]

[1]Collaborative Innovation Center on Forecast and Evaluation of Meteorological Disasters, Nanjing University of Information Science and Technology, Nanjing, China
[2]State Key Laboratory of Numerical Modeling for Atmospheric Sciences and Geophysical Fluid Dynamics, Institute of Atmospheric Physics, Chinese Academy of Sciences, Beijing, China
[3]Key Laboratory for Semi-Arid Climate Change of the Ministry of Education, College of Atmospheric Sciences, Lanzhou University, Lanzhou, China

*Corresponding author:* Tie Dai (daitie@mail.iap.ac.cn)

**Abstract.** The Cloud-Aerosol Transport System (CATS) lidar, on board the International Space Station (ISS), provides a new opportunity for studying aerosol vertical distributions, especially the diurnal variations from space observations. In this study, we investigate the seasonal variations and diurnal cycles of the vertical aerosol extinction coefficients (AECs) over East Asia by taking advantage of 32 months of the continuous and uniform aerosol measurements from the CATS lidar. Over the Tibetan Plateau, a belt of AECs approximately 6 km between 30°N and 38°N persistently exists in all seasons with an obviously seasonal variation. In summer, the aerosols at 6 km are identified as a mixture of both anthropogenic aerosols transported from India and coarse dust particles from Asian dust sources. In addition, the high AECs up to 8 km in summer over the Tibetan Plateau are caused by smoke aerosols from thermal dynamic processes. In fall and winter, the north slope of the plateau is continuously influenced by both dust aerosols and polluted aerosols transported upslope from the cities located in northwestern Asia at lower elevations. The diurnal variation of AECs in North China is mainly related to the diurnal variations of the transported dust and local polluted aerosols. Below 2 km, the AEC profiles in North China at 06:00 local Time (LT) and 12:00 LT are significantly higher than those at 00:00 LT and 18:00 LT, reaching the maximum at midday. The aerosol vertical profiles over the Tarim desert region in summer have obvious diurnal variations with the AECs at 12:00 LT and 18:00 LT being significantly higher than those at 00:00 LT and 06:00 LT, which are induced by the strongly diurnal variations in near-surface wind speeds. In addition, the peak of the AEC profiles has a significant seasonal variation, which is mainly determined by the boundary layer height.

## 1 Introduction

Knowledge of aerosol optical properties, such as aerosol optical thickness (AOT) and their vertical distributions is critical for estimating the effects of aerosol particles on air quality, radiative forcing and their related impacts on climate (Liu et al., 2011, 2014; Myhre et al., 2013; Ramanathan et al., 2001; Sato et al., 2018; 2019). Aerosol can significantly alter the vertical profile of solar heating, with great implications for atmospheric stability and dynamics within the lower troposphere (Huang



et al., 2009). However, investigation of aerosol particle properties and their temporal-spatial variations, especially their vertical structure, is still poor. The lack of information on the aerosol vertical distributions is one of the main underlying factors producing considerable uncertainty in aerosol direct radiative forcing, since the predictions from atmospheric models

typically suffer from great variability (Huneeus et al., 2011; Sekiyama et al., 2010; Yumimoto et al., 2008).

The lidar instrument is a useful tool in research on aerosol vertical distributions (Sugimoto and Huang, 2014); including aircraft-based lidar, ground-based lidar, and the space-based Cloud-Aerosol Lidar with Orthogonal Polarization (CALIOP) on board the Cloud-Aerosol Lidar and Infrared Pathfinder Satellite Observation (CALIPSO) satellite (Winker et al., 2007). Due to the geographic limitations of instruments, aircraft-based and ground-based observations are always carried out to

explore the climatology aspects of aerosol vertical structures and their associated climatic impacts at the selected sites (Liu et al., 2012; Huang et al., 2010). Although CALIOP can provide a better understanding of the effects of regional aerosols on radiative forcing (Jia et al., 2018; Oikawa et al., 2013, 2018), climate change (Huang et al., 2010), air quality (Huang et al., 2015; Yu et al., 2010), and cloud properties (Li et al., 2017) it is still difficult to investigate the full diurnal cycle of aerosol vertical structures by CALIOP because of its 16-day repeat cycle. Because aerosols can exhibit strong diurnal behavior,

observations of the aerosol diurnal variations are needed to improve simulations of pollution processes, geochemical cycles, and ultimately climate trends. The measurement of diurnal variations of aerosol vertical properties is especially crucial for visibility and aerosol particle forecasts.

The Cloud-Aerosol Transport System (CATS) lidar is installed on board the International Space Station (ISS) and observes with a nearly 3-day repeat cycle (McGill et al., 2015). It can provide a new opportunity for studying diurnal variations in

aerosol vertical distributions from space observations at the global scale. Compared with the lidar data from CALIOP, CATS can supply better temporal and spatial coverage of aerosols over the tropics and midlatitudes at different local times each overpass that are often hours apart from the CALIOP observations (York et al., 2016). Because CATS is a new lidar instrument, there have been relatively few studies conducted with CATS aerosol products to date. By comparing with the European Aerosol Research Lidar Network (EARLINET), Proestakis et al. (2019) presented a quantitative assessment of the

CATS Level 2 aerosol backscatter coefficient product and revealed that low negative biases may lead to the slight uncertainties of AOTs in climate studies. Lee et al. (2019) validated the CATS aerosol products using different multiplatform observations (i.e., the Aerosol Robotic Network (AERONET), Moderate Resolution Imaging Spectroradiometer (MODIS), and CALIOP) and analyzed the diurnal variations of aerosol extinctions from a global perspective. The results show that the CATS aerosol extinction profiles are in accordance with those from CALIOP despite

an apparent CALIOP underestimation in the lowest 2 km height. Rajapakshe et al. (2017) also reported differences between CATS and CALIOP in that the bottom of the above-cloud aerosol layer identified by CATS is much lower than that from CALIOP. In addition, Christian et al. (2019) applied the CATS data to evaluate the hemispheric transport of pyrocumulonimbus smoke aerosols simulated by a chemical transport model. Callewaert et al. (2019) also used the lidar measurements from CATS to conduct a qualitative comparison of dust aerosol concentration profiles retrieved by the

Mineral Aerosol Profiling from Infrared Radiances algorithm. The CATS products not only can catch the fast-moving

volcanic SO₂ and aerosol clouds, but they also constrain the trajectory-based estimates, thus producing more accurate dispersion patterns (Hughes et al., 2016). However, comprehensive research is still lacking that focuses on the East Asian aerosol features with different time scales based on the long-term and continuous satellite-based CATS observations, which specialize in providing high-frequency aerosol observations.

East Asia, with increasing air pollution during the last several decades, currently attracts considerable attention. The subregions over East Asia have different topographies and climates, and the atmospheric aerosols in East Asia are a complex mixture of various constituents including anthropogenic aerosols originating from complicated sources and the natural wind-blown dusts from desert regions. Thus, it is imperative to provide insights into the seasonal variations and diurnal cycles of the aerosol vertical features across East Asia.

Therefore, in this paper we present the results of analysis of temporal-spatial distributions of AOTs and the aerosol extinction coefficients (AECs) observed by CATS collected from 2015 to 2017 over East Asia. Brief descriptions of the selected regions and observations used in the analyses are introduced in Sect. 2. Section 3 shows the seasonal variations in CATS AOTs and compares them with those from MODIS to prove the reasonability of CATS aerosol products. The meridional cross-sections of AECs and aerosol depolarization ratios (ADRs), along with the seasonal and diurnal variations

of AEC profiles over three selected regions (North China, the Tibetan Plateau, and the Tarim Basin) are also presented in Sect. 3. Conclusions are given in Sect. 4.

## 2 Data and Methodology

### 2.1 Selected regions

The study domain was selected as a bounded region by 15-50°N and 70-140°E, as shown in Fig. 1. By considering the

differences in climate and terrain of each subregion of East Asia, we are not only studying East Asia as a whole but are also selecting typical regions based on the geographical location and dominant aerosol type for the comprehensive analysis of the aerosol optical properties over each region. Therefore, three regions (highlighted by red rectangles in Fig. 1) were selected: 1. the Tarim Basin, the largest dust source region in East Asia; 2. the Tibetan Plateau, the highest plateau in the world and an important moisture source affecting the global hydrological cycle; and 3. North China, the region of the great emissions of

anthropogenic aerosol pollution.

### 2.2 Datasets

### 2.2.1 CATS

The CATS (https://cats.gsfc.nasa.gov), launched in January 2015, is a multiwavelength lidar remote sensing instrument that provides vertical resolved measurements of clouds and aerosols from the ISS. The ISS orbit, which is at an altitude of

approximately 415 km and a 51° inclination, allows CATS to observe more comprehensive coverage of the tropics and midlatitudes at different local times each overpass with roughly a 4 day repeat cycle (McGill et al., 2015; York et al., 2016).



In this study, CATS Level 2 Operational (L2O) version 3.00 5 km aerosol profile products for the entire period of March 2015-October 2017 are used for analysis. CATS L2O aerosol profile products include day or night vertical profiles of geophysical parameters derived from Level 1 data, such as the vertical feature mask and profiles of aerosol properties (i.e.,

extinction, particle backscatter, and depolarization). CATS L2O aerosol profile data are provided at 1064 nm, with a uniform spatial resolution of 60 m vertically and 5 km horizontally over an altitude range of -1.0 to 30 km. CATS also provides data at 532 nm, but due to a laser-stabilization issue, 532 nm data are not recommended for use. We only use 1064 nm products, referring to York et al. (2016) and Lee et al. (2019).  In addition, owing to the limited contributions of aerosol above the troposphere, we only consider the data below an altitude of 10 km.

To identify the aerosol signals and remove uncertain observations, we conduct quality-control procedures for the aerosol extinction coefficients using several quality assurance thresholds (including an extinction quality control (QC) flag, feature type score and uncertainty of extinction coefficients) according to Lee et al. (2019). In addition, the total depolarization ratio is more constrained using a threshold to eliminate the missing value (Total_Depolarization_Ratio_1064_Fore_FOV >=0.0). After subjecting the data to quality control, we aggregate the original CATS aerosol extinction coefficients and total

depolarization ratios to 0.5° by 0.5° horizontal resolution and 0.24 km vertical resolution for each hour within ±30 minutes. The 0.5°×0.5° horizontal pixel size is provided for further aerosol data assimilation (Cheng et al., 2019).

**2.2.2 MODIS**

MODIS (https://modis.gsfc.nasa.gov) instruments on board both the Terra and Aqua platforms provide nearly daily global coverage of key atmospheric and land surface parameters (Remer et al., 2005; https://modis.gsfc.nasa.gov). The MODIS

Aqua and Terra Collection 6.1 Level 3 Dark Target (DT) and Deep Blue (DB) combined monthly AOTs at 550 nm from 2015-2017 are averaged as the climatological values for CATS AOTs validation.

**3 Results**

**3.1 Seasonal variations of AOTs in East Asia**

Figure 2 shows the spatial distributions of CATS AOTs at 1064 nm and MODIS AOTs at 550 nm for the four seasons

(spring: MAM; summer: JJA; fall: SON; winter: DJF), during the period of March 2015-October 2017. To construct Fig. 2, quality-assured CATS aerosol extinction coefficients are first binned on a 0.5°×0.5° grid over the globe and then vertically integrated within 10 km altitude as CATS AOTs. Overall, although MODIS AOTs are systematically higher than CATS AOTs over East Asia, both CATS and MODIS AOTs yield similar spatial patterns of heavy aerosol loadings clustered in Northwestern China, Eastern China and Northern India. The biases between MODIS and CATS AOTs are probably not only

because of the instrument capabilities in different channels but also the deficiency of lidar systems in detecting tenuous layers of signal below the minimum detection thresholds (Proestakis et al., 2019). Moreover, the MODIS aerosol products monitor the ambient aerosol properties over cloud-free conditions (Levy et al., 2013), whereas the aerosol optical properties in CATS are retrieved in all sky conditions. In Northwestern China, the locations of hot spots with high AOTs remain



unchanged throughout the whole year, but the strengths of AOT centers have significant seasonal variations. Both the CATS
and MODIS AOTs caused by the natural dust aerosols mobilized around the Taklimakan Desert are higher than 0.5 in spring
and summer, and decrease to 0.3 in fall and winter. The dust storms driving into Northwest China from the Taklimakan
Desert, combined with those from the Gobi Desert could also enhance the AOTs over the downwind regions in China,
especially in springtime (Shao et al., 2011). The significant seasonal variations in the high AOT regions over Eastern China
are found in MODIS AOTs depending on the seasonal changes of aerosol emissions and influences of the East Asian
monsoon (Li et al., 2016; Wu et al., 2015). The anthropogenic emissions of sulfate and carbonaceous aerosols in Eastern
China are related to the seasonal cycle of human activities including industry, agriculture and transportation. Compared with
those during the cold season (spring and winter), the MODIS AOTs in the Sichuan Basin are obviously reduced in the warm
season (summer and autumn). This is probably due to the strong wet removal of aerosols by a sufficient warm and wet flow
from the Pacific Ocean. Due to the intense emissions from biomass burning in spring over Southeast Asia, a mass of aerosols
is transported from the source regions to South China, causing the significantly high MODIS AOTs in spring. Over Eastern
China, the continuous high MODIS AOTs are located in the Beijing-Tianjin-Hebei Urban Agglomeration throughout the
year and peak in the summer season. In summer, convective turbulence may transport low-level aerosol upward to a high
altitude, and high humidity and temperature conditions can increase the rate of gas-particle transformation and hygroscopic
growth, which will increase aerosol backward scattering, and thus lead to the high AOT in the areas with relatively higher
pollutant emissions (Henriksson et al., 2011). In fall and winter, the northern cold, dry and clean winter monsoon accelerates
the diffusion of aerosols, thus decreasing the AOTs in Eastern China. However, since 1064 nm measurements are less
sensitive to fine-mode aerosols such as smoke and pollutant aerosols compared to coarse aerosols such as dust aerosols, the
CATS has some difficulty catching the seasonal variations of AOTs in Eastern China, which are controlled by fine particles.
For the entire year, the CATS AOTs in Eastern China are generally approximately 0.3, whereas the MODIS AOTs are more
than twice those of the CATS, particularly during spring and summer. The annual cycle of AOT in India consists of
superimposed seasonal cycles of fine anthropogenic particles and coarse natural particles. Therefore, the seasonal cycles in
India are strong, with the seasonal cycle for anthropogenic aerosols having its maximum in the winter and that of natural
aerosols having its maximum in summer (Henriksson et al., 2011). The major natural aerosol over South Asia is the wind-
blown mineral dust from the arid and semiarid regions of southwest Asia, such as the Thar Desert. Both CATS and MODIS
can catch the maximums of AOTs around northwestern India in summer caused by dust aerosols due to the higher wind
speeds during this season.

Similar to Fig. 2, Fig. S1 shows the spatial distribution of CATS AOTs that are aerosol extinction columns integrated below
1 km, from 1 to 2 km and above 2 km from the ground for the four seasons. Fig. S2 is similar to Fig. S1, but shows the total
depolarization ratios. The annual spatial patterns of AOTs below 1 km and between 1 and 2 km are generally in accordance
with that of the column-integrated AOTs below 10 km. Above 2 km, it is difficult to find any characteristics in the spatial
distributions, which is probably because of the scarce aerosols and the complicated transport pathways above the boundary
layer. This indicates that the spatial distributions of column aerosols mainly rely on the near-surface aerosols rather than





long-range transport. In contrast to the slight AOT variations in horizontal orientation above 2 km, the annual total depolarization ratios above 2 km show an obvious decreasing tendency from Northwest to Southeast Asia, which is also
revealed below 2 km. This means that a part of the dust aerosols with relatively small size are lifted from the source regions by vertical convection and transported in the troposphere. In summer, the mean integrated AOTs above 1 km are significantly higher than those below 1 km in the Taklamakan Desert, which reports stronger heat convection in summer than in spring over this source region. Moreover, it is interesting to find that the AOTs in Northeast China in spring are obviously higher than in other seasons and most of the aerosols are gathered above 2 km, which is also found by Qiu et al.
(2018). According to the frequency of occurrences of the CATS-derived vertical aerosol subtypes in Northeast China, the aerosols above 2 km in spring are controlled by dust and dust mixture aerosols, which are mainly transported by the westerlies from the Asian dust source regions.

### 3.2 Meridional cross-sections of AECs and ADRs over selected regions

In this section, the aerosol vertical distributions are studied as the meridional cross-sections for the three selected regions in
the four seasons. The probability distributions of the CATS-derived vertical aerosol subtypes in Fig. 3 are shown as further analysis of the discrepancies of aerosol compositions in different regions. Fig. 4 and Fig. 5 give the vertical structure of AECs and ADRs observed from CATS by season over North China, the Tibetan Plateau, and the Tarim Basin in relation to the zonal mean regional surface elevation. In addition, to explore the diurnal variations of AECs meridional cross-sections, Figs. S3-S8 show the 6 h (00:00, 06:00, 12:00, 18:00 LT) vertical structure of AECs and ADRs observed from CATS by
season over these three selected regions. Dust aerosols have a large linear depolarization ratio due to the nonsphericity of dust particles, which is different from other types of aerosols. Therefore, the ADR is an effective parameter for the identification of dust aerosols (Murayama et al., 2001). In this section, a value of 0.1 is chosen as the threshold of ADR for identifying dust aerosols according to Liu et al. (2012).

### 3.2.1 North China

Our statistics show that North China is a region with complicated aerosol compositions and the dominant compositions vary with season and height. As shown in Fig. 3, the dominant aerosol in North China is pure dust aerosol in all seasons, and the subdominant aerosol in spring, summer, fall, and winter is smoke (19.18%), smoke (27.41%), smoke (20.20%), and polluted continental aerosols (15.91%), respectively. Polluted continental aerosols strongly impact the aerosol vertical distribution below 2 km in all the seasons except spring. Throughout the year, the intraseasonal variations in the planetary boundary
layer in winter are significantly weaker than in other seasons. The probability distribution of polluted continental aerosols always peaks approximately 1 km and the height of the peak is slightly shifted following the variations in boundary layer height. This is probably due to the thermal inversion layer near the surface preventing the upward diffusion of aerosols and most of the aerosols emitted from surface can only be concentrated at the top of boundary layer. The peak of pure dust occurrences is always slightly higher than that of mixed dust aerosols. This illustrates that the mixed dust in North China is
mostly generated by mixing with the aerosol particles from near-surface emissions rather than by aerosol transport above the



boundary layer. The smoke aerosols are generally concentrated within 2-4 km. This indicates that external input controls the smoke aerosols in this region and their primary transport path is obviously higher than that of dust aerosols.

Due to the region having high AECs concentrated in the boundary layer whereas the pattern of ADRs is totally different from that of the AECs, it is obvious that fine particles control the aerosols near the surface. Although the AECs in spring are generally smaller than 0.1 km$^{-1}$ above the boundary layer, the ADRs with values higher than 0.16 still peak at approximately 2 km. This indicates that the dust aerosols with relatively large size emitted from the Asian desert are suspended as the dominant aerosol type at this height in spring. Only in spring, the AECs in 42-46°N are greater than 0.09 km$^{-1}$, which is probably related to the elevated smoke and reflects harvest seasons in Northeast China. Moreover, due to the high ADRs in this region, dust aerosol is also one of the important aerosol types in Northeast China through its long-range transport from the Gobi Desert. Compared with the AEC and ADR over Northeast China in spring, the lower AEC and comparable ADR in summer show the reduced agricultural activities and the continuous influences of dust storms. A region of high AEC values exists throughout the whole year between 32 and 38°N, which includes the Beijing-Tianjin-Hebei Urban Agglomeration with frequent human activities and heavy industrial production. The height of AECs greater than 0.1 km$^{-1}$ in summer is obviously higher than in other seasons, reflecting the combined effects of the local aerosol emissions and stronger vertical diffusion. The patterns of AEC and ADR cross-sections in North China are generally similar in fall and winter, apart from the region with ADRs higher than 0.08 between 32 and 36°N above 2 km in winter. This is probably due to the limited dust aerosols transported from remote areas by strong northwest winds during this season.

The AECs and ADRs in North China have significantly diurnal variations (Figs. S3-S6). In all seasons, the AECs at 00:00 LT are lowest and peak at 12:00 LT with strongly vertical transport especially between 32 and 38°N. The lowest ADRs are also at 00:00 LT, which indicates that the dominant aerosols at 00:00 LT are fine particles. In spring, the ADRs with values greater than 0.2 reach as high as 4 km at 06:00 LT, and the ADRs at 12:00 LT and 18:00 LT are significantly higher than at 12:00 LT in the boundary layer between 32 and 38°N. This indicates that the dust aerosols being transported from sources in spring always arrive over this region at 06:00 LT and sink by gravitational processes in the daytime. The ADRs at 06:00 LT, 12:00 LT, and 18:00 LT are significantly higher than that at 00:00 LT in spring and summer, while the ADRs at 00:00 LT and 06:00 LT are comparable in fall and winter. This further proves that human activities control the diurnal variations of aerosols in fall and winter rather than the dust storms that are the dominating factor in spring and summer.

### 3.2.2 Tibetan Plateau

The Tibetan Plateau is located at the juncture of several important natural and anthropogenic aerosol sources. The Tibetan Plateau is located at the junction of several dust sources, including the Taklimakan desert, Gurbantunggut desert, and Great Indian Thar desert, and all have important effects on the plateau in different atmospheric layers. The boundary layer height in the Tibetan Plateau varies slightly by season, whereas the intraseasonal variations in the boundary layer are comparable in all seasons. As shown in Fig. 3, the dominant aerosol subtype in the Tibetan Plateau in the column is dust aerosol (Huang et al., 2007), followed by smoke aerosol. The pure dust (smoke) aerosols account for 66.22% (14.14%), 60.66% (18.17%), 53.50% (18.21%), and 42.20% (25.02%) in spring, summer, fall, and winter, respectively. In agreement with the CALIPSO




results in Huang et al. (2007), the probability distributions of dust aerosol in all seasons almost peak approximately 6 km, which are contributed by the transported dust aerosols from the nearby Taklamakan Desert surface. In fall and winter, almost 74.64% and 78.54% of polluted continental aerosols are concentrated below 2 km, which are the most dominant aerosol subtypes near the surface.

As shown in Fig. 4 and Fig. 5, it is obvious that the dust emitted from the Taklimakan Desert could be transported

southwardly to the Tibetan Plateau. The belt region approximately 6 km between 30°N and 38°N in the Tibetan Plateau prevails in all seasons and the AECs and ADRs in this region have significantly seasonal variations. The belt region has the highest AECs and ADRs in spring and there is a significant trajectory to show the transport of dust aerosols from the Taklimakan Desert through the north slopes of the Tibetan Plateau after the topographic lifting (Liu et al., 2019b). The transported dust stacking up against the slopes of the Tibetan Plateau can heat up the elevated surface air over the slopes by

absorbing solar radiation, and then further cause a large-scale circulation anomaly (Lau et al., 2006; Liu et al., 2008). It also affects the convective clouds over the Tibetan Plateau and then causing heavy rainfall in the downstream region (Liu et al., 2019b; 2020). Considering summer, the diffusion of aerosols is stronger than other seasons and therefore the AECs greater than 0.05 km$^{-1}$ are as high as 8 km, which is caused by smoke aerosols through thermal dynamic processes (Fig. 3). It is noted that although the value of AECs at approximately 6 km in summer is comparable to that in spring, the aerosol sources

are totally different between these two seasons. As shown in Fig. 4 and Fig. 5, due to the distinct demarcations approximately 29°N in spring, we supposed that there is limited aerosol lifted from Northern India and transported to the mountains. However, in summer, the AECs at 6 km in the Tibetan Plateau are homogeneous and the ADRs in the south of 30°N and north of 30°N are generally smaller and greater than 0.1, respectively. This shows that the aerosols at 6 km in summer are a mixture of both fine particles such as anthropogenic aerosols lifted and transported from northern India, and

coarse dust particles from dust sources in Asia. In fall and winter, the ADR to the north of 37°N are slightly lower than that above the mountains, which indicates that the north slope of the plateau is continuously influenced by not only dust aerosol but also polluted aerosols transported upslope from the cities located in northwestern Asia at lower elevations. On the south slope of the Tibetan Plateau, the aerosols are mainly from the desert in northwestern India and the urban cites with frequent human activities. The proportion of smoke aerosols in fall and winter is higher than other latitude zones between 30°N and

32°N, implying strong local emissions of smoke in this region.

Due to the inconsistent effects of aerosol factor on cloud properties between daytime and nighttime (Liu et al., 2019a), the diurnal variations of aerosols have a dominant role in influencing the cloud microphysical processes over the Tibetan Plateau. The AECs and ADRs above the mountains have distinct diurnal variations in spring (Figs. S4-S7). AECs at 12:00 LT and 18:00 LT are generally higher than those at 00:00 LT and 06:00 LT whereas the ADRs at 12:00 LT and 18:00 LT above

mountain are significantly lower than those at 00:00 LT and 06:00 LT. This indicates the southwardly transported coarse dust aerosols from the Tarim Basin always arrive at the Himalayan mountains during nighttime and the fine particles related to residential heating constitute the dominant aerosols in the daytime during spring. It is interesting that the mixture of aerosols from the south and north slopes approximately 29°N are more intense during nighttime than daytime. In the warm





season (summer and fall), AECs above the mountains are comparable throughout the day although ADRs show a decreasing

trend after sunrise.

### 3.2.3 Tarim Basin

As shown in Fig. 3, most of the layers in Tarim Basin are dominated by pure dust aerosols. The column-integrated occurrences of pure dust aerosols are 86.77%, 79.47%, 78.95%, and 65.31% in spring, summer, fall, and winter, respectively. The peak of dust occurrence has a significant seasonal variation and reaches its highest altitude in summer, which is in

accordance with the changes in boundary layer height in the Tarim Basin. Strong convective activity resulted in the height of the pure dust aerosol probability peak of 3-4 km in summer; however, the height of the peak is trapped below 1 to 2 km by subsidence in winter. It is interesting that the season with weaker intraseasonal variabilities of boundary layer has more squeezed probability distributions of dust and polluted continental aerosols. This indicates that boundary layer height is an important factor determining the vertical distributions of aerosols from local emissions in the Tarim Basin. The frequency of

smoke aerosols always peaks approximately 4 km in all the seasons with the unimodal structure, indicating the smoke aerosols in this region have a great possibility originating from a distance and generally being transported at 4 km altitude.

As expected, Fig. 4 shows that AECs in the Tarim Basin have significantly seasonal variations. Because of the uniform aerosol component in the Tarim Basin, the tempo-spatial variation of AECs is generally in agreement with that of ADRs in Fig. 5. It should be noted that the dust activities over this area appear to be persistent almost all year long, reaching a

maximum in spring and a minimum in winter. Although most of the aerosol layers are concentrated within the boundary layer and change with the variations in boundary layer height (peaking in summer), the dust layers can be found as high as 5 km altitude in all seasons except winter. Such separation leads to the creation of a well-defined aerosol vertical structure that can be transported over long distances. Throughout the whole year, the dust aerosols emitted from the dust source region below 2 km are generally transported to Central Asia by the East Wind and the dust aerosols lifting as high as 2 km are

spread eastwards (Huang et al., 2008). The AECs in spring are significantly higher than those in summer whereas on the contrary the ADRs in summer are much higher and peak at 2 km above surface. This indicates that a significant amount of dust aerosols with larger sizes is lifted into the free troposphere and suspended for a longer time in summer than that in spring. The geographic setting of the basin surrounded by high mountains generates atmospheric circulations in the basin that are favorable for dust to remain suspended for a long time in the air (Tsunematsu et al., 2005). In Fig. 4, the zonal means

indicate that intense dust events are generally confined between 38°N and 42°N, which is in agreement with the CALIPSO results in Liu et al. (2008).

The AECs over Tarim Basin have slightly diurnal variations in spring (Figs. S5-S8), whereas the ADRs at 00:00 LT and 06:00 LT are significantly higher than at 12:00 LT and 18:00 LT. This is because the near-surface atmosphere becomes stable during nighttime and thus blocks the diffusion of large particles. In summer, AECs reveal obviously diurnal variations,

and the AECs at 12:00 LT and 18:00 LT are higher than those at 00:00 LT and 06:00 LT. In fall and winter, the high AECs are generally near the surface, especially at 19:00 LT, which is related to frequent residential activities.

### 3.3 Seasonal and diurnal variations of AEC profiles over selected regions





Using quality-assured CATS-derived aerosol vertical distributions, the annual mean CATS extinction vertical profiles at 1064 nm by season over North China, the Tibetan Plateau, and the Tarim Basin are shown in Fig. 6. The AECs in North

China have the weakest seasonal variations among these three regions; moreover, the differences in intraseasonal AEC variations among the four seasons are also very limited and generally proportional to the values of the AECs. North China has AECs in spring and summer that are slightly higher than those in fall and winter. Through all the seasons, AECs in North China decline rapidly with increasing altitude below 2 km and level out above 3 km. The differences in AEC profiles over North China in different seasons exist below 0.5 km and between 1.5 to 4 km, which is related to the seasonal variations of

local pollutant emissions and the long-range transport of aerosols above the boundary layer. The AEC profiles in the Tibetan Plateau are significantly discrepant among the four seasons. The AEC in the cold season (spring and winter) generally deceases monotonically following the increase in altitude below 4 km and peaks at approximately 5.5 km with AECs less than 0.1 km $^{-1}$. In contrast, the AEC profile in the warm season (summer and fall) has a similar bimodal pattern and first peaks at 1.17 and 0.93 km, and then once more peaks at approximately 6 km. The intraseasonal variabilities of AECs over

the Tibetan Plateau are more prominent in spring and summer than in fall and winter over most of the layers, which is probably caused by the frequent dust events from adjacent areas with different strengths. In the Tarim Basin, the AEC profiles in all seasons except winter are unimodal and generally peak around the altitude of the dust sources. Between 2.5 and 4 km, the AEC in summer is homogeneously distributed at various altitudes while the AEC in spring and fall is reduced significantly with the increase in altitude. The intraseasonal variabilities of AECs over the Tarim Basin below 2 km are more

intense in winter than in other seasons. This is probably because the residential coal combustion in cities located at lower elevations has obviously diurnal variations and thus causes the significant changes in AECs below 2 km.

As shown in Fig. 7, the annual mean AEC vertical profiles at 1064 nm observed from CATS are binned into 00:00, 06:00, 12:00, and 18:00 local times (LT) based on the closest match in time to give the diurnal variations of AECs over North China, the Tibetan Plateau, and the Tarim Basin. The heights of the regional mean planetary boundary layer every 6 h (00:00,

06:00, 12:00, 18:00 UTC) are also given in Fig. 7. For the North China region, the AEC profiles in the four seasons all present a declining trend with height. Among the four seasons, the differences in AEC profiles between the four local times are more remarkable in spring, summer, and fall than in winter. Except in winter, the AEC profiles below 2 km at 06:00 LT and 12:00 LT are significantly higher than those at 00:00 LT and 18:00 LT, and reach their maximum at midday. This indicates the aerosols in the boundary layer are mostly emitted in the daytime heating period and the vertical mixing of

aerosols is enhanced by the combined role of boundary layer and thermal wind circulation. Following the diurnal variations in the boundary layer, AEC profiles show the strongest aerosol diffusions with the highest boundary layer height at 06:00 UTC. For the Tibetan Plateau, the rate of decline of AECs with altitude is higher than that in the other two regions. In spring, the AEC profiles at 06:00 LT, 12:00 LT, and 18:00 LT are monotonically decreasing, while the AEC profile at 00:00 LT first increases and then decreases with its maximum at the 1-2 km layer. In the warm season, due to the mixture of various

aerosols originating from different regions above the mountains, the AEC profiles at approximately 6 km have minor diurnal variations. In winter, the AEC profile at 00:00 LT is the lowest among the four local times. Between 5 and 7 km, the AECs



at 00:00 LT and 06:00 LT are lower than those at 12:00 LT and 18:00 LT, indicating that the input of aerosols and local sources in winter above the mountains are more frequent in the daytime. For the Tarim Basin, dust aerosol is prevalent throughout the year. To investigate the main reasons for the diurnal variations in the AEC profiles in the Tarim Basin, we

give the annual mean 10 m wind fields in the four local times by season in Fig. 8. Both in spring and summer, the AEC profiles at the four times have a similar unimodal pattern. In spring, the AEC profiles are consistent in each local time with the maximum aerosol extinctions in the 1-4 km altitude range. This indicates that dust emissions over the Tarim Basin in spring have minor diurnal variations. Interestingly, the AEC profiles in summer have strong diurnal variations and the AECs at 12:00 LT and 18:00 LT are significantly higher than those at 00:00 LT and 06:00 LT. As shown in Fig. 8, the regional

annual mean wind speeds in the Tarim Basin are 1.94 m/s, 2.15 m/s, 2.64 m/s, and 2.09 m/s at 00:00 LT, 06:00 LT, 12:00 LT, and 18:00 LT, respectively. Because the dust emissions are highly correlated to the surface wind field (Dai et al., 2018), the obviously diurnal variations in the AEC profile are mainly attributed to the diurnal variations in the surface wind speeds in summer. In fall and winter, the AEC profiles in the four local times all have a maximum below 2 km, and peak at 12:00 LT (18:00 LT) in fall (winter).

**4 Summary and Conclusions**

In this paper, we took advantage of the variable local time of overpass of the International Space Station to document the seasonal variation and diurnal cycle of the aerosol vertical profile as seen by the CATS lidar. Our results are based on 32 months of systematic observations collected during 2015-2017, which enable statistically significant results.

We first evaluated the seasonal variations of AOTs over East Asia from CATS against those from MODIS and found

satisfactory agreement on the spatial patterns. The positive biases between MODIS and CATS AOTs are probably due to the differences in channels and the deficiency of lidar systems in detecting tenuous layers of signal below the minimum detection thresholds. Retrieving under different cloud conditions can also result in discrepancies between MODIS and CATS aerosol products. Moreover, CATS has some difficulty in catching the seasonal variations of AOTs in the regions controlled by anthropogenic aerosols.

Due to the differences in aerosol compositions of each subregion over East Asia, we selected three typical regions (North China, the Tibetan Plateau, and the Tarim Basin) for further investigations into aerosol vertical features with different time scales. The dominant aerosol in North China is dust aerosol in all seasons and the subdominant aerosol in spring, summer, fall, and winter is smoke (19.18%), smoke (27.41%), smoke (20.20%), and polluted continental aerosols (15.91%), respectively. In North China, human activities control the diurnal variations of aerosols in fall and winter whereas the dust

storm is the dominant factor in spring and summer. The most dominant aerosol subtype in the Tibetan Plateau is dust aerosol, followed by smoke aerosol. In addition, polluted continental aerosol controlled the aerosols below 2 km in fall and winter. The belt regions approximately 6 km between 30°N and 38°N in the Tibetan Plateau exist in all seasons and the AECs and ADRs in this region have significantly seasonal variations. The belt region has the highest AECs and ADRs in spring and



there is a significant trajectory showing the transport of dust aerosols through the north slopes of the Tibetan Plateau after the
topographic lifting. The aerosols at 6 km in summer are a mixture of both anthropogenic aerosols transported from northern
India and coarse dust particles from Asian dust sources. The north slope of the plateau in fall and winter is continuously
influenced by both dust aerosols and polluted aerosols transported upslope from the cities located in northwestern Asia at
lower elevations. The high AECs in summer over the Tibetan Plateau up to 8 km are caused by smoke aerosols. The coarse
aerosols transported southwards from the Tarim Basin always arrive at the Himalayan mountains during nighttime and the
fine particles related to residential heating constitute the dominant aerosols in daytime in spring. The Tarim Basin is
dominated by dust aerosols throughout the year, and the AECs have significantly seasonal variations. Throughout the whole
year, dust aerosols emitted from the dust source region below 2 km are generally transported to Central Asia by the East
Wind and the dust aerosols lifting as high as 2 km are spread eastwards. Compared with spring, a significant amount of dust
aerosols with larger sizes is lifted into the free troposphere and suspended for a longer time in summer.

The AECs in North China have the weakest seasonal variations among these three regions. The intraseasonal variabilities of
AECs over the Tibetan Plateau are more prominent in spring and summer than in fall and winter over most of the layers,
which is probably caused by the frequent dust events from adjacent areas with different strengths. In the Tarim Basin, AEC
profiles in all seasons except winter are unimodal and generally peak around the altitude of the dust sources. The intensely
intraseasonal variabilities of AECs over the Tarim Basin below 2 km in winter is due to the obviously diurnal variations of
residential coal combustion in the cities that are located at lower elevations.

The diurnal variations of AECs in North China are mainly related to the diurnal variations of the transported dust and local
polluted aerosols. Below 2 km, the AEC profiles at 06:00 LT and 12:00 LT are significantly higher than those at 00:00 LT
and 18:00 LT, and reach their maximum at midday. The highest AECs at 12:00 LT approximately 6 km in the Tibetan
Plateau indicate the aerosols emitted from different sources are more easily uplifted and spread southwards at midday. Due
to the strongly diurnal variations of surface wind speed in summer, the AEC profiles over Tarim Basin have strong diurnal
variations and the AECs at 12:00 LT and 18:00 LT are significantly higher than those at 00:00 LT and 06:00 LT.

**Code/Data availability.** The data and data analysis method are available upon request.

**Author contributions.** Tie Dai and Jiming Li designed the study. Yueming Cheng conducted the data analysis with
contributions from all coauthors. Yueming Cheng prepared the manuscript with help from Tie Dai, Jiming Li, and Guangyu
Shi.

**Competing interests.** The authors declare that they have no conflict of interest.

**Acknowledgments.** This research has been supported by the Strategic Priority Research Program of the Chinese Academy
of Sciences (grant no. XDA2006010302), the National Key R&D Program of China (grant nos. 2016YFC0202001 and
2017YFC0209803), and the National Natural Science Funds of China (grant nos. 41571130024, 41605083, 41590875, and





41475031). We are grateful to the relevant researchers who provided the observation data from MODIS (https://modis-atmos.gsfc.nasa.gov/products/aerosol). CATS data were obtained from the NASA Langley Research Center Atmospheric Science Data Center (https://cats.gsfc.nasa.gov).



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





**Figures**

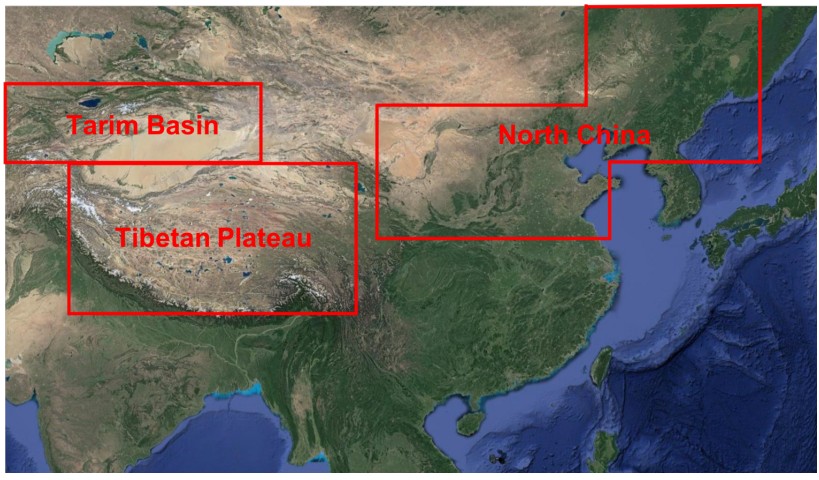


**Figure 1. The selected three regions used in this study. The map was created by © Google Earth (version: © Google Earth Map 2020).**



**Figure 2. Seasonally averaged AOT distributions from CATS at 1064 nm and MODIS at 550 nm for 2015-2017. (a,b) Spring (MAM); (c,d) summer (JJA); (e,f) fall (SON); (g,h) winter (DJF).**



**Figure 3. Probability distributions (frequency of occurrence) of the CATS-derived vertical aerosol subtypes by season over the three regions. The grey lines and fill areas correspond to the regional mean planetary boundary layer (km) and one standard deviation of the seasonal average, respectively.**



**Figure 4. Vertical structure of 1064 nm aerosol extinction coefficients (km⁻¹) observed from CATS by season over the three regions in relation to zonal mean regional surface altitude. The dotted line represents the zonal mean regional planetary boundary layer (PBL).**







**Figure 5. Vertical structure of total depolarization ratios observed from CATS by season over the three regions in relation to zonal mean regional surface altitude. The dotted line represents the zonal mean regional planetary boundary layer (PBL).**





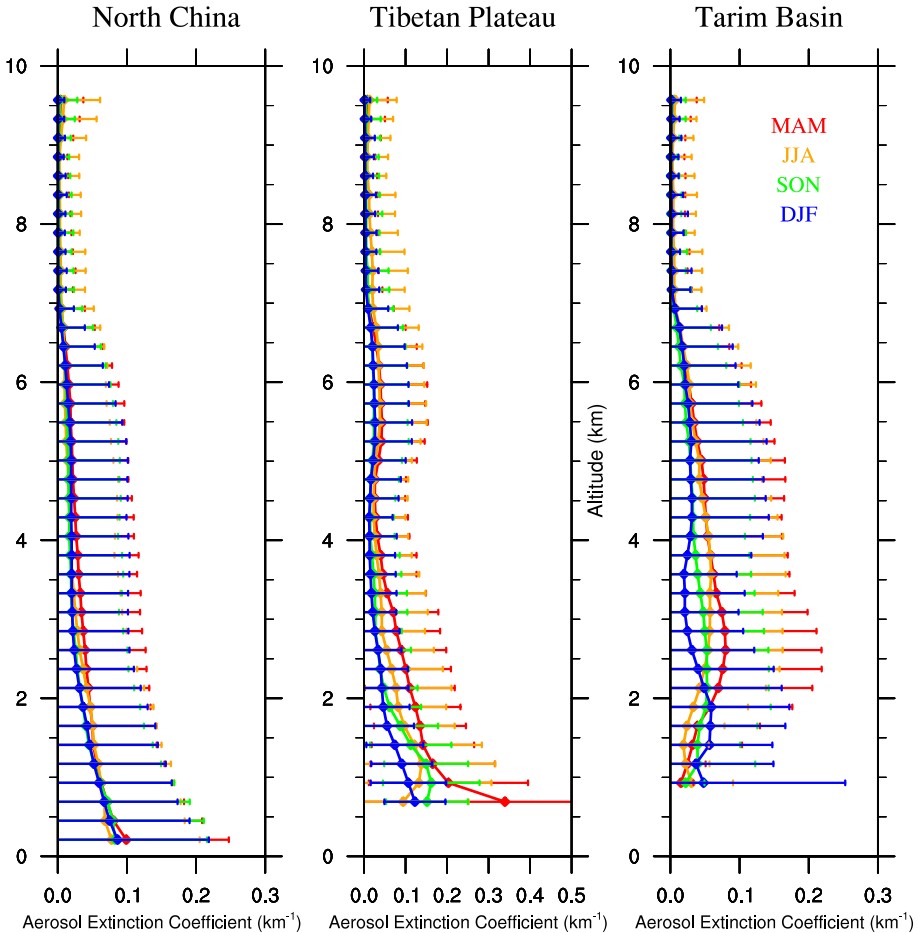

**Figure 6.** Vertical profiles of 1064 nm aerosol extinction coefficients (km⁻¹) observed from CATS by season over the three regions. Horizontal lines correspond to the standard deviation of the intraseasonal aerosol extinctions.

Figure 7. 6 h (00:00, 06:00, 12:00, 18:00 LT) vertical profiles of 1064 nm aerosol extinction coefficients (km⁻¹) observed from CATS by season over the three regions. The dashed lines represent the regional mean planetary boundary layer (PBL) by every 6 h (00:00, 06:00, 12:00, 18:00 UTC).




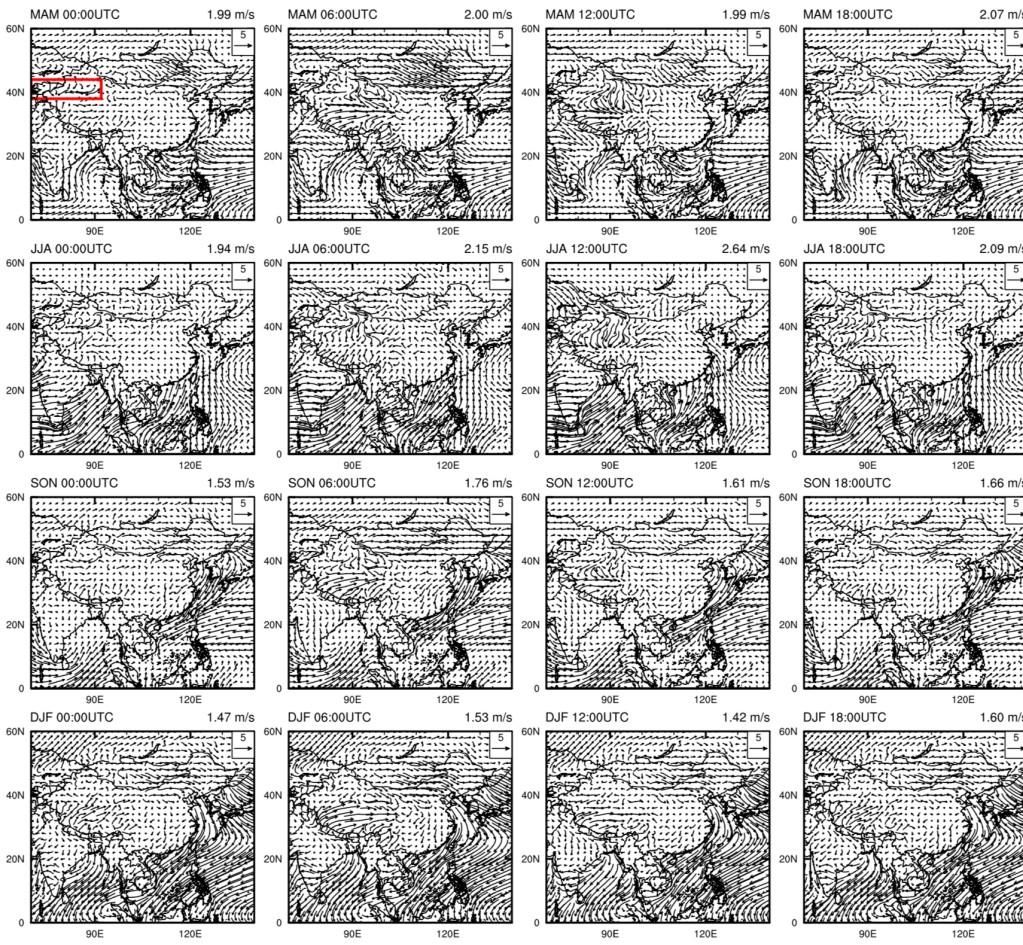

**Figure 8. Spatial distributions of the annual mean 10 m wind fields by every 6 h from NCEP FNL reanalysis data. The value in the top right corner represents the regional mean 10 m wind speed in the red rectangle.**