# Peer review of "Measurement Report: Determination of aerosol vertical features on different time-scales over East Asia based on CATS aerosol products"

_Atmospheric Chemistry and Physics, 2020_

## Referee Comment (RC1) · Anonymous Referee #1 · 6 Sep 2020

General comments:

The aerosol vertical distribution is crucial to the study of aerosol climate and environmental effects. Characterizing the diurnal variation of aerosol species in the atmosphere on a regional basis is an important project and can only be achieved by satellite measurements. Currently, due to the orbit limitations in most satellite instruments, it is still challenging for people to have a complete acknowledge of aerosol vertical structure. Moreover, there is seldom study focusing on the diurnal variations of aerosol vertical distribution. In this paper, the authors take the advantage of the Cloud-Aerosol Transport System (CATS) lidar on board the International Space Station (ISS) to report

the temporal-spatial distributions of aerosol properties especially the diurnal vertical variations over the East Asia. This paper also investigates the possible reasons for the aerosol vertical variations in three typical regions. This measurement report is a good contribution to our understanding of the aerosol vertical features and the aerosol diurnal variations over East Asia. It is also helpful for the improving of aerosol vertical modelling over this important region. Generally speaking, the manuscript is scientific sound and well written and organized. I recommend to accepting it after minor revision.

Major comments:

1. The number of CATS observations over each selected regions are significantly different during the four local times. The authors should clearly present the available sample of the observations per 6-hour and discuss the sample effect on the analysis of aerosol diurnal variations.

2. Although CALIPSO is unable to provide the aerosol observations at various local times, since it is the longest existing satellite with lidar system, the authors should consider to validate the CATS observations using the CALIOP observations at least during the overpass times.

Specific comments:

1. Lines 95-96, it is difficult to understand 'allows CATS to observe more comprehensive coverage of the tropics and midlatitudes at different local times each overpass with roughly a 4 day repeat cycle', please restructure this sentence.

2. Line 107, please clarify the quality-control procedures in detail.

3. Restructure the sentence in Lines 139-140.

4. Figure 2, 'Spring' should start with lower case in figure caption.

5. Lines 185-186, the authors state that 'the dominant compositions vary with season and height' and 'the dominant aerosol in North China is pure dust aerosol in all seasons'. There are some inconsistencies between these two sentences, please modify it.

6. Figure 3, the color of 'Marine Mixture' is difficult to distinguish.

---

## Referee Comment (RC2) · Anonymous Referee #2 · 2 Oct 2020

The vertical profile of aerosol is important for aerosol forecast, assimilation and pollution control. But the measurement information of vertical structure is still poor, especially in the West China, since there are few ground-based lidars. In this manuscript, the authors use the CATS lidar to investigate the seasonal variations and diurnal cycles of the vertical aerosol extinction coefficients. The results are helpful for the understanding of the vertical structure of aerosol. I recommend this paper for publication after the following points are addressed. 1. In Line 49 and Line 96, the repeat cycle of the CATS measurements described is different, please check it carefully. 2. In Line 108, Please introduce the detail of the total depolarization ratios, for the readers to understand the related results. 3. In Line 127, Does CATS AOT contain aerosol information below the

[Figure]

cloud? If yes, it should be described in detail. 4. In Line 175, Does the CATS-derived aerosol subtypes product directly provided by CATS? Does it represent the total mass concentration of aerosol species in full particle size? 5. In Line 187, the pdf values are 19.18, 27.41%, 20.20%, but the maximum value of x coordinate in Fig. 3 are less than 8%. 6. In Fig. 3, the maximum proportion of some species is above the top of the PBLH. How the PBLH and mean PBLH is calculated? Please describe it in detail. And the PBLH should be compared with some literatures. 7. I think the dust storms are usually rare in summer for more precipitation. Please explain the continue influence of the dust storms in summer. 8. If the authors consider the difference of local time among different time zones even in a region, it's suggested that the authors introduce the method of data matching.
* * *

---

## Author Comment (AC1) · 3 Nov 2020

**Response to the Comments of Referees**

**Measurement Report: Determination of aerosol vertical features on different time-scales over East Asia based on CATS aerosol products**

Yueming Cheng, Tie Dai, Jiming Li, Guangyu Shi

We would like to thank to the two reviewers for giving constructive criticisms, which are very helpful in improving the quality of the manuscript. We have made minor revision based on the critical comments and suggestions of the referees. The referee's comments are reproduced (black) along with our replies (blue) and changes made to the text (red) in the revised manuscript. All the authors have read the revised manuscript and agreed with submission in its revised form.

**Anonymous Referee #1**

**Comment NO.1:** *The aerosol vertical distribution is crucial to the study of aerosol climate and environmental effects. Characterizing the diurnal variation of aerosol species in the atmosphere on a regional basis is an important project and can only be achieved by satellite measurements. Currently, due to the orbit limitations in most satellite instruments, it is still challenging for people to have a complete acknowledge of aerosol vertical structure. Moreover, there is seldom study focusing on the diurnal variations of aerosol vertical distribution. In this paper, the authors take the advantage of the Cloud-Aerosol Transport System (CATS) lidar on board the International Space Station (ISS) to report the temporal-spatial distributions of aerosol properties especially the diurnal vertical variations over the East Asia. This paper also investigates the possible reasons for the aerosol vertical variations in three typical regions. This measurement report is a good contribution to our understanding of the aerosol vertical features and the aerosol diurnal variations over East Asia. It is also helpful for the improving of aerosol vertical modelling over this important region. Generally speaking, the manuscript is scientific sound and well written and organized. I recommend to accepting it after minor revision.*

**Response:** We thank the referee for this very positive assessment of our manuscript.

**Comment NO.2:** *The number of CATS observations over each selected regions are*

*significantly different during the four local times. The authors should clearly present the available sample of the observations per 6-hour and discuss the sample effect on the analysis of aerosol diurnal variations.*

**Response:** Accept. Fig. S9 shows the sample fractions of the available aerosol extinction coefficients observed from CATS by the four China standard times (CST) for the whole of one day. Over North China, the fractions of the CATS observations among the four local times are equivalent at most altitude except above 7 km where the diurnal variations of aerosol extinctions are limited. Over Tibetan Plateau and Tarim Basin, the number differences of CATS observations between the four times are generally concentrated below 2 km and above 7 km. Due to the significant discrepancies of observation number among the four China standard times below 2 km in these two regions, the confidence of the diurnal variations near surface is lower than at 2-6 km.

[Figure]

Figure S9. 6 h (00:00, 06:00, 12:00, 18:00 CST) sample fractions of the available aerosol extinction coefficients observed from CATS by season over the three regions for the whole of one day.

**Changes in Manuscript:** Please refer to the revised manuscript, Page 11 Line 409-416 and Fig. S9.

**Comment NO.3:** *Although CALIPSO is unable to provide the aerosol observations at various local times, since it is the longest existing satellite with lidar system, the authors should consider to validate the CATS observations using the CALIOP observations at least during the overpass times.*

**Response:** Accept.

Cloud–Aerosol Lidar and Infrared Pathfinder Satellite Observation (CALIPSO), as part of the A-train, was launched on April 2006 (Winker et al., 2007). All the satellites of the A-train are in a 705 km sun-synchronous polar orbit with an equator-crossing time at the local time of about 1:30 p.m. and a 16-day repeat cycle. The Cloud-Aerosol Lidar with Orthogonal Polarization (CALIOP), which carried by CALIPSO, is a spaceborne two-wavelength (532 nm and 1064 nm) polarization lidar that has been acquiring high-resolution vertical and geographical information of clouds and aerosols over global scales and providing an unprecedented opportunities to study the dust aerosols over bright surfaces and beneath thin clouds (Huang et al., 2007; 2009). Here, we use the aerosol extinction coefficients at 1064 nm in the CALIPSO lidar level 2 version 4.10 aerosol profile products below 10 km altitude (http://www-calipso.larc.nasa.gov/). The original aerosol extinctions have been operated with several quality assurance procedures and then aggregated to the 0.5°×0.5° horizontal resolution and 0.24 km vertical resolution for each hour within ±30 minutes. The detailed of the CALIOP profile data processing is shown in Cheng et al. (2019).

As CALIPSO is the longest existing satellite with lidar system, the CATS-derived aerosol extinction vertical distributions are cross-compared against collocated CALIOP aerosol extinction vertical distributions in 2017 as the validations of the CATS aerosol extinctions. In order to guarantee the spatio-temporal consistencies of observations from CALIOP and CATS, the collocated profiles for CALIOP and CATS are averaged only when the observations from the two sensors are same in space and time. It should be noted that the differences in algorithm

and instrument can lead to differences in extinction coefficients.

In Fig. A1, the annual mean profiles of the normalized aerosol extinction coefficients observed from CALIOP and CATS at 1064 nm over East Asia are generally consistent in magnitude. Both these two aerosol profiles peak around 0.5 km and show decreasing trend with height above 0.7 km. It indicates that the averaged vertical distributions of aerosols from CATS compare reasonably well with that from CALIOP.

[Figure]

Figure A1. Vertical profiles of annual mean normalized aerosol extinction coefficients observed from CALIOP and CATS at 1064 nm over East Asia in 2017.

References:

Cheng, Y., Dai, T., Goto, D., Schutgens, N. A. J., Shi, G. and Nakajima, T.: Investigating the assimilation of CALIPSO global aerosol vertical observations using a four-dimensional ensemble Kalman filter, Atmos. Chem. Phys., 19(21), 13445–13467, doi:10.5194/acp-19-13445-2019, 2019.

Huang, J., Minnis, P., Yi, Y., Tang, Q., Wang, X., Hu, Y., Liu, Z., Ayers, K., Trepte, C. and Winker, D.: Summer dust aerosols detected from CALIPSO over the Tibetan Plateau, Geophys. Res. Lett., 34(18), L18805, doi:10.1029/2007GL029938, 2007.

Huang, J., Minnis, P., Chen, B., Huang, Z., Liu, Z., Zhao, Q., Yi, Y. and Ayers, J. K.: Long-range transport and vertical structure of Asian dust from CALIPSO and surface measurements during PACDEX, J. Geophys. Res., 113(D23), D23212, doi:10.1029/2008JD010620, 2008.

Huang, J., Fu, Q., Su, J., Tang, Q., Minnis, P., Hu, Y., Yi, Y., and Zhao, Q.: Taklimakan dust aerosol radiative heating derived from CALIPSO observations using the Fu-Liou radiation model with CERES constraints, Atmos. Chem. Phys., 9, 4011–4021, https://doi.org/10.5194/acp-9-4011-2009, 2009.

Winker, D. M., Hunt, W. H. and McGill, M. J.: Initial performance assessment of CALIOP, Geophysical Research Letters, 34(19), doi:10.1029/2007GL030135, 2007.

**Changes in Manuscript:** Please refer to the revised manuscript, Page 4 Line 123-124.

**Comment NO.4:** *Lines 95-96, it is difficult to understand 'allows CATS to observe more comprehensive coverage of the tropics and midlatitudes at different local times each overpass with roughly a 4 day repeat cycle', please restructure this sentence.*

**Response:** Agree. We have restructured this sentence as 'One benefit of CATS is the ISS orbital characteristic with 3 day repeat cycle, which enables better temporal and spatial coverage of measurement over the tropics and midlatitudes than that of CALIOP'.

**Changes in Manuscript:** Please refer to the revised manuscript, Page 3 Line 111-112.

**Comment NO.5:** *Line 107, please clarify the quality-control procedures in detail.*

**Response:** Accept. The quality-control procedures include the following: (1) the extinction QC value should be equal to 0, indicating the final lidar ratio is unchanged and non-opaque layer; (2) the feature type must be determined as aerosols only; (3) the score of feature type should be greater than -10 and lower than -2, indicating high confidence in discriminating aerosols; (4) the uncertainty of the extinction coefficient must be lower than 10 km$^{-1}$, indicating a stable iteration.

**Changes in Manuscript:** Please refer to the revised manuscript, Page 4 Line 147-151.

**Comment NO.6:** *Restructure the sentence in Lines 139-140.*

**Response:** Agree. We have restructured this sentence as 'Due to the intense biomass burning

emissions in spring over Southeast Asia, biomass burning aerosols transported from source regions cause the significantly high MODIS AOTs over South China'.

**Changes in Manuscript:** Please refer to the revised manuscript, Page 5 Line 189-191.

**Comment NO.7:** *Figure 2, 'Spring' should start with lower case in figure caption.*

**Response:** Agree.

**Changes in Manuscript:** We have replaced 'Spring' with 'spring', please refer to the caption of Fig. 2.

**Comment NO.8:** *Lines 185-186, the authors state that 'the dominant compositions vary with season and height' and 'the dominant aerosol in North China is pure dust aerosol in all seasons'. There are some inconsistencies between these two sentences, please modify it.*

**Response:** Agree. We have replaced 'the dominant aerosol in North China is pure dust aerosol in all seasons' with 'the dominant aerosol type of column integrated aerosol in North China is pure dust aerosol in all seasons'.

**Changes in Manuscript:** Please refer to the revised manuscript, Page 7 Line 248-249.

**Comment NO.9:** *Figure 3, the color of 'Marine Mixture' is difficult to distinguish.*

**Response:** Agree. We have replaced the color of 'Marine Mixture'.

**Changes in Manuscript:** Please refer to Fig. 3.

**Anonymous Referee #2**

**Comment NO.1:** *The vertical profile of aerosol is important for aerosol forecast, assimilation and pollution control. But the measurement information of vertical structure is still poor, especially in the West China, since there are few ground-based lidars. In this manuscript, the authors use the CATS lidar to investigate the seasonal variations and diurnal cycles of the vertical aerosol extinction coefficients. The results are helpful for the understanding of the vertical structure of aerosol. I recommend this paper for publication after the following points are addressed.*

**Response:** We thank the referee for this very positive assessment of our manuscript.

**Comment NO.2:** *In Line 49 and Line 96, the repeat cycle of the CATS measurements described is different, please check it carefully.*

**Response:** Agree.

**Changes in Manuscript:** We have replaced the "4 day repeat cycle" with "3 day repeat cycle" in Line 96 to make it be consistent with the repeat cycle in Line 49. Please refer to the revised manuscript, Page 3 Line 111-112.

**Comment NO.3:** *In Line 108, Please introduce the detail of the total depolarization ratios, for the readers to understand the related results.*

**Response:** The linear volume total depolarization ratio is defined as the ratio of perpendicular total (Rayleigh plus particle) backscatter to parallel total backscatter. The value of the depolarization ratio depends on the symmetry of the molecule and the normal vibrational mode. Total depolarization ratios are reported for each 5 km profile and 60 m range bin in which atmospheric particulates were detected. Dust aerosols have a large linear depolarization ratio due to the non-sphericity of dust particles, which is different from other types of aerosols. Therefore, the ADR is an effective parameter for the identification of dust aerosols (Murayama et al., 2001). In this study, 0.25 and 0.15 are chosen as the thresholds of ADR for classifying dust and dust mixture aerosols.

Reference:

Murayama, T., Sugimoto, N., Uno, I., Kinoshita, K., Aoki, K., Hagiwara, N., Liu, Z., Matsui, I., Sakai, T., Shibata, T., Arao, K., Sohn, B.-J., Won, J.-G., Yoon, S.-C., Li, T., Zhou, J., Hu, H., Abo, M., Iokibe, K., Koga, R. and Iwasaka, Y.: Ground-based network observation of Asian dust events of April 1998 in east Asia, J. Geophys. Res., 106(D16), 18345–18359, doi:10.1029/2000JD900554, 2001.

**Changes in Manuscript:** We have added the detail of the total depolarization ratios in the revised manuscript. Please refer to the revised manuscript, Page 4 Line 136-143.

**Comment NO.4:** *In Line 127, Does CATS AOT contain aerosol information below the cloud? If yes, it should be described in detail.*

**Response**: Yes, CATS AOT can contain aerosol information below the cloud. The aerosol optical depths of CATS are obtained by integrating the 1064 nm aerosol extinction profile. If there are clouds in the column that are found have horizontal oriented ice (HOI) crystals, it is likely that the quality of the column optical depth is low. The anomalously high backscatter from HOI clouds generally makes the extinction retrieval more difficult. Because all the data below the HOI cloud is rescaled by the retrieval optical depth, the extinction data below could

be suspect.

**Changes in Manuscript:** Please refer to the revised manuscript, Page 4 Line 124-129.

**Comment NO.5:** *In Line 175, Does the CATS-derived aerosol subtypes product directly provided by CATS? Does it represent the total mass concentration of aerosol species in full particle size?*

**Response:** Yes, the CATS-derived aerosol subtypes product is directly provided by CATS, and it does not represent the total mass concentration of aerosol species in full particle size. For each atmospheric layer defined as an aerosol in the feature type parameter, an assessment of the aerosol type is reported for each 5 km profile and 60 m range bin in which atmospheric particulate layers were detected. The aerosol features for CATS are retrieved depending on the feature integrated depolarization ratio at 1064 nm, feature integrated total attenuated backscatter at 1064 nm, surface type (for maritime) and feature altitude, which have heritage from CALIOP aerosol typing algorithm (Omar et al., 2009). Moreover, the Modern-Era Retrospective analysis for Research and Applications, version 2 (MERRA-2) aerosol products are incorporated to help guide aerosol typing in instances where the observed quantities are characteristic of multiple aerosol types.

Reference:

Omar, Ali H., and coauthors, 2009: The CALIPSO Automated Aerosol Classification and Lidar Ratio Selection Algorithm, J. Atmos. Oceanic Technol., 26, 1994–2014.

**Changes in Manuscript:** We have added the descriptions of the CATS-derived aerosol subtypes product in the revised manuscript. Please refer to the revised manuscript, Page 4 Line 129-135.

**Comment NO.6:** *In Line 187, the pdf values are 19.18, 27.41%, 20.20%, but the maximum value of x coordinate in Fig. 3 are less than 8%.*

**Response:** This pdf value represents the integrations of the pdf value at 0.24 km range bin below 10 km altitude.

**Comment NO.7:** *In Fig. 3, the maximum proportion of some species is above the top of the PBLH. How the PBLH and mean PBLH is calculated? Please describe it in detail. And the PBLH should be compared with some literatures.*

**Response:** Agree. the maximum proportion of some species is above the top of the PBLH. The

PBLH every 6 h is from the National Centers for Environmental Prediction (NCEP) Final (FNL) Analysis (NOAA/NCEP, 2000) (https://rda.ucar.edu/datasets/ds083.2/). This PBLH represents the height of the atmospheric layer from the Earth's surface up to an altitude of about 1 kilometer in which wind speed and direction are affected by frictional interaction with objects on the Earth's surface. The mean PBLH is simply averaged and the standard deviation of PBLH indicates the dispersion of the PBLH in each season for each region.

Reference:

NOAA/NCEP, 2000. NCEP FNL Operational Model Global Tropospheric Analyses, Continuing from July 1999 (Updated Daily). National Center for Atmospheric Research Computational and Information Systems Laboratory Research Data Archive. https://rda.ucar.edu/datasets/ds083.2/.

**Changes in Manuscript:** We have introduced the PBLH reanalysis data in the revised manuscript. Please refer to the revised manuscript, Page 6-7 Line 234-240.

**Comment NO.8:** *I think the dust storms are usually rare in summer for more precipitation. Please explain the continue influence of the dust storms in summer.*

**Response:** Agree. The precipitation over the eastern part of northern China in summer is significantly greater than that in other seasons and dust occurrence decreased dramatically over the eastern part of northern China in summer because precipitation and vegetation limit the area of dust lifting and enhance dust wet deposition (Wang et al., 2013). The prior precipitation also restrains the occurrence of sand dust storms in Taklamakan Desert (Xiao et al., 2008). However, due to the annual precipitation is small and strong summer convection favors dust lifting, the dust storms reached maxima in spring and minima in winter in Taklamakan Desert. Dusty days in dust-frequent years were associated with strong wind days when the precipitation is about 10 mm and dust occurrences were suppressed by large amounts of precipitation (approximately 22 mm) in dust-less years over the southeastern part of the Gobi Desert (Amgalan et al., 2017). Because dust occurs at low altitudes and is mostly limited to below a height of 2 km in fall and winter, the dust occurrence over the northeast of the Tibetan Plateau is significantly higher in spring and summer (Wang et al., 2013).

In summer, the dust plumes originate from the nearby Taklamakan desert surface and accumulate over the northern slopes of the Tibetan Plateau. And these dust outbreaks can affect

the radiation balance of the atmosphere of Tibet because they both absorb and reflect solar radiation (Huang et al., 2007).

Reference:

Amgalan, G., Liu, G.-R., Kuo, T.-H. and Lin, T.-H.: Correlation between dust events in Mongolia and surface wind and precipitation, Terr. Atmos. Ocean. Sci., 28(1), 23–32, doi:10.3319/TAO.2016.04.25.01(CCA), 2017.

Huang, J., Minnis, P., Yi, Y., Tang, Q., Wang, X., Hu, Y., Liu, Z., Ayers, K., Trepte, C. and Winker, D.: Summer dust aerosols detected from CALIPSO over the Tibetan Plateau, Geophys. Res. Lett., 34(18), L18805, doi:10.1029/2007GL029938, 2007.

Wang, H., Zhang, L., Cao, X., Zhang, Z. and Liang, J.: A-Train satellite measurements of dust aerosol distributions over northern China, Journal of Quantitative Spectroscopy and Radiative Transfer, 122, 170–179, doi:10.1016/j.jqsrt.2012.08.011, 2013.

Xiao, F., Zhou, C. and Liao, Y.: Dust storms evolution in Taklimakan Desert and its correlation with climatic parameters, J. Geogr. Sci., 18(4), 415–424, doi:10.1007/s11442-008-0415-8, 2008.

**Comment NO.9:** *If the authors consider the difference of local time among different time zones even in a region, it's suggested that the authors introduce the method of data matching.*

**Response:** Despite the three selected regions spanning several geographical times, we hypothesize that the China standard time (CST) (offset of UTC+08:00) is used as the local time in all the three regions, because the largest difference between the local time in the three regions and CST is almost two hours.

**Changes in Manuscript:** We have replaced the 'LT' with 'CST' to avoid ambiguity. Please refer to the revised manuscript.

---

## Author Comment (AC2) · 3 Nov 2020

The comment was uploaded in the form of a supplement:
https://acp.copernicus.org/preprints/acp-2020-715/acp-2020-715-AC2-supplement.pdf